# OpenReview forum: "ExID: Offline RL with Intuitive Expert Insights in Limited-Data Settings"
_NeurIPS.cc/2024/Conference — Submitted to NeurIPS 2024_

### Official Review · Reviewer_UWfE · 2024-07-10

**Soundness:** 3
**Presentation:** 2
**Contribution:** 3
**Rating:** 6
**Confidence:** 3

**Summary:**

The paper introduces ExID, an offline reinforcement learning algorithm that enhances learning performance in limited data scenarios by combining domain knowledge in the form of simple decision trees with agent experience replay data.

**Strengths:**

* Domain Knowledge Utilization: ExID incorporates domain knowledge to guide decision-making in data-limited scenarios
* Teacher-Student Architecture: A teacher network, informed by domain knowledge, regularizes a student critic network to improve generalization.
* Regularization with Domain Knowledge: The algorithm uses a regularization term to align the critic's decisions with the teacher's advice for states covered by domain knowledge.

**Weaknesses:**

* Discrete Action Space Limitation: The algorithm is currently limited to discrete action spaces, necessitating future extensions for continuous action domains.
* Hyperparameter Tuning Challenge: The need for precise hyperparameter tuning complicates the deployment of ExID in scenarios where extensive optimization is impractical.
* The paper does not have enough strong experiment comparisions. The methods of the paper is related with offline RL methods, such as SCQ[1], ReDS[2], A2PR[3], CPED[4]. But it lacks the experiments comparisions with offlien RL methods. I think adding some SOTA baseline methods will improve your paper. It is not required that experimental comparisons must be given, but at least add some discussion with these methods to the paper.

References：

[1] Shimizu, Yutaka, et al. "Strategically Conservative Q-Learning." arXiv preprint arXiv:2406.04534 (2024).

[2] Singh, Anikait, et al. "ReDS: offline reinforcement learning with heteroskedastic datasets via support constraints." Proceedings of the 37th International Conference on Neural Information Processing Systems. 2023.

[3] Liu, Tenglong, et al. "Adaptive Advantage-Guided Policy Regularization for Offline Reinforcement Learning." In International Conference on Machine Learning (ICML). PMLR, 2024.

[4] Zhang, Jing, et al. "Constrained policy optimization with explicit behavior density for offline reinforcement learning." Advances in Neural Information Processing Systems. 2023

**Questions:**

* How does ExID perform when the true optimal policy deviates significantly from the provided domain knowledge? Can the algorithm recover from incorrect domain knowledge without additional corrective mechanisms?
* What measures are in place to ensure robustness when the domain knowledge contains errors or biases? How sensitive is the algorithm to such inaccuracies?
* The paper mentions a real sales promotion dataset, but how does ExID perform in other real-world scenarios? Are there any plans for more extensive real-world testing to validate the algorithm's practical utility?
* How should practitioners select the hyperparameters $\lambda$ and $k$ in the absence of extensive computational resources?

**Limitations:**

* The paper only conducts experiments in several simulated environments and a real-world sales promotion dataset, which may not fully verify the effectiveness and applicability of the algorithm in more diverse and complex real-world scenarios.
* The performance of the ExID algorithm heavily relies on the quality of the domain knowledge. If the domain knowledge is incomplete, inaccurate, or biased, it may mislead the learning process and result in suboptimal policies. Moreover, obtaining high-quality domain knowledge can be challenging and time-consuming in practice.
* The proposed method mainly concentrates on discrete action spaces, and its performance and applicability in continuous action spaces are not clear. This limits the algorithm's utility in many real-world control tasks that involve continuous action spaces.

---

> ### Author Rebuttal · Authors · 2024-08-06
>
> Thank you for your valuable feedback. Please find our response below:
>
> **W1: Discrete Action Space Limitation**
>
> - We conducted experiments on discrete action space domains as many important real world problems use discrete action policies. For example navigation with actions forward or turning, finance trading with actions buy, hold or sell etc. We have also mentioned the use of discrete policies in our limitations.
> - However, we would like to highlight the proposed methodology can be extended to any continuous domain problem by using the regularization in Eq 4 during critic $(Q_s^\theta)$ training for continuous domain and using actions from actor network $(\pi_s)$ for cross entropy loss in Eq 7.
>  - We empirically demonstrate this by conducting additional experiment by training policy on continuous version of sales promotion task (action range coupon (0 - 5), discount (0.6 - 1) ) and  continuous Type-1 Diabetes basal bolus control task (action range basal and bolus (-1 to 1) (results in Q3).
>
> | Environment |CQL+$\mathcal{D}$ |SCQ |EXID|Performance gain|
> |--|--|--|--|--|
> | SP|679.25$\pm$ 35.02 | 708.44 $\pm$ 52.19 |827.76 $\pm$ 43.79  | 14.38% |
>
> **W2 and Q4: Hyperparameter Tuning Challenge**
>
> - We agree hyperparameter tuning in offline setting is challenging. In practice this can be done by using tuning methods like Bayesian Optimization and observing the behavior of Q values as proposed by [2].
> - $k$ and $\lambda$ should be selected with peak Q value observed during training. We discuss the effect of different $k$ and $\lambda$ in Fig 11 and Fig 12 (supplement) and empirically show $\lambda$ = 0.5 and $k$ = 30 works well in most setting. We also show the proposed methodology is better than $\lambda$ = 0 and $k = 0$ demonstrating robustness to hyperparameters.
> - The Q based tuning does not require additional computational resource.
>
> [2] Kumar, A., Singh, A., Tian, S., Finn, C., & Levine, S. (2021). A workflow for offline model-free robotic reinforcement learning. arXiv preprint arXiv:2109.10813.
>
> **W3: Strong experiment comparisons**
>
> - The paper compares ExID to SOTA discrete offline RL methods. However continuous control algorithms like SCQ, ReDS, A2PR, CPED also suffer from performance degradation for OOD state. We empirically compare EXID to SCQ for SalesPromotion continuous  task. Please find the results in table of W1.
> - The suggested offline RL methods use action constraints to correct OOD actions for states in dataset but do not have any mechanism for states unseen in dataset. It has been recently established generalization is the main bottleneck of offline RL [3].
> - ExID addresses this generalization gap by using knowledge distillation and teacher network updates to correct actions for states not seen during training. We will add this discussion in our revised manuscript.
>
> [3] Park, Seohong, et al. "Is Value Learning Really the Main Bottleneck in Offline RL?." _arXiv preprint arXiv:2406.09329_ (2024).
>
> **Q1: Deviation and Recovery from incorrect Domain Knowledge**
>
> - In practice it is never possible to obtain perfect domain knowledge. Thus the algorithm updates the initial teacher network ($\pi_t^\omega$) obtained from domain knowledge using Eq 5, 6 and 7 during training.
> - Thus ExID uses a self correction methodology to find the optimal policy even when there is significant deviation from the domain knowledge. Please refer to Pg 5 for details of the teacher update.
> - The domain knowledge used for the experiments is also not optimal as reported in performance of behavior cloned teacher under the column $\mathcal{D}$ of Table 1 and 2.
>
> **Q2: Robustness to errors and biases in domain knowledge.**
>
> - The self correction methodology provides robustness to the errors and biases in domain knowledge. We discuss the Effect of varying $\mathcal{D}$ quality in Sec 5.6. Fig 6 a shows using imperfect knowledge (Rule 3) and knowledge with high error (Rule 2) with ExID can bring substantial performance improvement over baseline.
> - We agree using absolutely incorrect domain knowledge will cause performance degradation as shown in Fig 6 a and mentioned in limitations. However, domain knowledge is available for practical problems in domains like business and healthcare.
>
> **Q3: Other real world scenarios**
>
> We conducted experiments on 6 benchmark and open source datasets for reproducibility. The method can be applied to many real problems in business, healthcare where domain knowledge is available. However such datasets are often proprietary. If the reviewer could kindly suggest any other opensource real world dataset we would be happy to conduct experiments. Additionally we tested ExID for Type 1 diabetes control task by administrating basal and bolus insulin level [2]. *This is a continuous control task*
> - The following basic basal bolus insulin control is known for diabetic patients
>
> Domain knowledge :
> 1. basal = u2ss * BW / 6000 where
> u2ss: The patient's or default steady-state insulin.
> BW: The patient's or default body weight.
> 2. meal > 0 $\implies$ bolus = (carbohydrate / carbohydrate_ratio) + (current_glucose - target_glucose) / correction_factor)/ sample_time
>
> The offline data is obtained from open access NeoRL2 github repository.
>
> | Environment |$\mathcal{D}$| CQL+$\mathcal{D}$ |EXID|Performance gain|
> |--|--|--|--|--|
> | SimGlucose |17.53 $\pm$ 3.02 | 21.79 $\pm$ 3.60 |30.82 $\pm$ 6.95  | 41.44% |
>
> [2]. Jinyu Xie. Simglucose v0.2.1 (2018) [Online].
>
> **Q4: Hyperparameter selection**
>
> Please refer to our response of weakness 1.
>
> **Limitation : Dependence on high quality domain knowledge**
>
> We would like to highlight the domain knowledge used in this work is suboptimal (obtained from heuristic) and incomplete (covering only part of state space). This can be observed in Table 1 and 2 under $\mathcal{D}$ where performance of the initial domain knowledge is not optimal. Heuristic based domain knowledge is available for many real world use cases for business, healthcare etc.

---

> > ### Comment · Reviewer_UWfE · 2024-08-11
> > **Official Comment by Reviewer UWfE**
> >
> > Thanks for your clear explanation and careful answers, which have addressed my concern. I see you add the comparison experiments with SCQ. I expect you to add more comparison experiments or discussions with other SOTA offline RL methods ReDS, A2PR, and CPED as mentioned earlier in your paper. Then, I am willing to raise my score.

---

> > > ### Author Response · Authors · 2024-08-11
> > > **Additional baselines**
> > >
> > > We thank the reviewer for acknowledging the rebuttal and for raising important questions that strengthen our paper. Please find the below comparison with mentioned offline RL methods:
> > >
> > > **ReDS: offline reinforcement learning with heteroskedastic datasets via support constraints**
> > >
> > > The main contribution of RedS is to provided distribution constraints against a reweighted version of the behavior policy. This facilitates the policy to freely choose state-by-state how much the learned policy should stay close to the behavior policy for states seen in the the dataset.  This method is applicable to the datasets showing heteroskedastic distributions (different behavior actions for the same states) whereas our methodology employs action correction for  OOD states that are not observed in the dataset. Since our dataset is not heteroskedastic this methodology is not directly comparable with ExID.
> > >
> > > **Adaptive Advantage-Guided Policy Regularization for Offline Reinforcement Learning**
> > >
> > > A2PR trains a VAE similar to SCQ to identify high advantage that differ from those present in the dataset. The VAE is trained with $\log p{\psi}(a|s) \geq \mathbb{E}{q_{\phi}(z|a,s)} \left[ \mathbb{1}\{f(A(s,a)) > \epsilon_A\} \log p_{\psi}(a|z,s) \right] - \text{KL}\left[q{\phi}(z|a,s) \parallel p(z|s)\right]$ where $s \in \mathcal{B}r$. This method does not estimate actions for $s \notin {B}r$ which ExID does via knowledge distillation.
> > >
> > >
> > > **Constrained policy optimization with explicit behavior density for offline reinforcement learning**
> > >
> > > CPED uses a flow-GAN model to explicitly estimate the density of behavior policy. This facilitates choosing different actions which are safe for the for states in dataset. The flow GAN model is trained on the dataset generated by behavior policy and does not account for the states outside the dataset.
> > >
> > > Please find the comparison of the baselines with ExID in the table below:
> > >
> > > | Environment | CQL+$\mathcal{D}$ | SCQ | A2PR | CPED | EXID |
> > > |-------------|------------------|-----|------|------|------|
> > > | SP          | 679.25$\pm$ 35.02 | 708.44 $\pm$ 52.19 | 712 $\pm$ 32.09 | 715 $\pm$ 47.31 | 827.76 $\pm$ 43.79 |
> > >
> > >
> > > In summary all these methods do not employ any action correction mechanism for OOD states outside the dataset leading to performance degradation in case of limited data. As a result these algorithms perform almost similarly on sales promotion dataset. ExID distills knowledge for OOD states from domain knowledge leading to performance enhancement over the baseline methods. We add this discussion to the revised manuscript.
> > >
> > >  If there are any further questions we will be happy to answer them. Otherwise we will be grateful if you could reconsider your score.

---

> > > > ### Comment · Reviewer_UWfE · 2024-08-11
> > > > **Official Comment by Reviewer UWfE**
> > > >
> > > > Thanks for your careful discussion, I have improved the score from 4 to 6.

---

> > > > > ### Author Response · Authors · 2024-08-11
> > > > > **Thank you for discussion**
> > > > >
> > > > > Dear Reviewer UWfE,
> > > > >
> > > > > Thanks for you for carefully reading the response and for helping us improve the paper! We will incorporate the additional comparisons in the revised version of the manuscript.
> > > > >
> > > > > Thank you again for your valuable comments and guidance.
> > > > >
> > > > > Best,
> > > > >
> > > > > Authors

---

### Official Review · Reviewer_NyjZ · 2024-07-11

**Soundness:** 3
**Presentation:** 2
**Contribution:** 2
**Rating:** 6
**Confidence:** 4

**Summary:**

This paper studies offline RL when data is limited. The authors propose a domain knowledge-based regularization technique to learn from an initial tracker network and limited data buffer. The experiments verified the effectiveness of the proposal, which outperforms the classic RL baseline methods.

**Strengths:**

1. The proposed method is simple and technically reasonable.
2. The experimental results on the real sales promotion dataset show the proposal is a promising solution in real-world applications.

**Weaknesses:**

1. The technical novelty is limited. Despite the claimed use of expert knowledge, the method adopted by the paper is to directly train a policy from the knowledge, which assumes that the information provided by the domain knowledge is at the state-action level (a decision tree in this paper), which limits the feasibility of this method. Compared to the use of knowledge between latent concepts discussed in neuro-symbolic learning, I think it's more like traditional model distillation.
2. In practice, limited offline data may come from domain knowledge-based strategies, such as human-designed rules, thus I have great concerns about whether these two can promote each other. Empirical studies on more real-world datasets or rigorous theoretical analysis will provide support to this issue and further improve this work.
3. The introduction uses the sales task as an example, but the visualization is based on the Mountain Car dataset.
4. Definition 4.1 seems strange, why not directly define the offline dataset as a subset of the complete state spaces?
5. The $\eta$ in Proposition 4.2 is not well defined.

**Questions:**

1. Could the “Intuitive Expert Insights” be formally defined?
2. The difference between the proposal and the traditional model distillation.


---

After rebuttal, my concerns have been addressed. I decide to raise the score to 6.

**Limitations:**

The authors have provided a discussion about the limitations.

---

> ### Author Rebuttal · Authors · 2024-08-06
>
> Thank you for your valuable feedback. Please find our response below:
>
> **W1: Limited novelty and comparison with traditional knowledge distillation**
>
> - The domain knowledge considered in our setting is imperfect and is updated using expected improvement of RL policy in a completely offline manner using Eq 6 and 7 which is different from traditional knowledge distillation methods. This update is essential as shown by the ablation study in Fig 5 c.
> - The study of performance degradation of offline RL due to limited data is unexplored in literature as agreed upon by Reviewer Cy8u. To the best of our knowledge, a knowledge distillation based regularization approach has not been studied for Out of Distribution (OOD) action correction for unseen states in dataset (OOD states) in offline RL prior to this work.
> -  Please refer to discussion in Pg 18 Section E "Related Work: Knowledge Distillation" for comparison with prevalent knowledge distillation methods in RL.
>
> **W2 : Offline data from domain knowledge, empirical studies on more real-world datasets or rigorous theoretical analysis**
>
> - In practice offline data generally comes from human demonstrations. Since it is costly to explore all possible states during data collection, the offline data distribution is often narrow. The policy learned on such demonstrations fails on encountering OOD states. Please refer to Pg 16 Fig 8 for pictorial representation of the complementarity of domain knowledge to the dataset.
> - The policy performance is improved over these OOD states via incorporation of domain knowledge. Proposition 4.2 formalizes this notion with respect state coverage and $\mathcal{D}$ quality theoretically establishing performance improvement is possible as highlighted by reviewer Cy8u. Please refer to pg 13 App A for the full proof.
> - We conducted experiments on 6 standard datasets. The results of Minigrid environment are in Table 3 pg 19 due to page limitation in the main manuscript. Additionally we tested ExID for diabetes management task by administrating basal and bolus insulin level [2]. *This is a continuous control task*.
> - The following basic basal bolus insulin control is known for diabetic patients
>
> Domain knowledge :
> 1. The basal insulin is based on the insulin amount to keep the blood glucose in the steady state when there is no (meal) disturbance.
>
> basal = u2ss (pmol/(L*kg)) * body_weight (kg) / 6000 (U/min)
>
> 2. The bolus amount is computed based on the current glucose level, the target glucose level, the patient's correction factor and the patient’s carbohydrate ratio.
>
> bolus = ((carbohydrate / carbohydrate_ratio) + (current_glucose - target_glucose) / correction_factor)/ sample_time
>
> The offline data is obtained from open access NeoRL2 github repository.
>
> | Environment |$\mathcal{D}$| CQL+$\mathcal{D}$ |EXID|Performance gain|
> |--|--|--|--|--|
> | SimGlucose |17.53 $\pm$ 3.02 | 21.79 $\pm$ 3.60 |30.82 $\pm$ 6.95  | 41.44% |
>
> -  It has been recently established generalization is the main bottleneck of offline RL [1]. ExID addresses this generalization gap by using knowledge distillation and teacher network updates to correct actions for states not seen during training as shown in Fig 4 Pg 8.
>
> [1] Park, Seohong, et al. "Is Value Learning Really the Main Bottleneck in Offline RL?." _arXiv preprint arXiv:2406.09329_ (2024).
> [2]. Jinyu Xie. Simglucose v0.2.1 (2018) [Online].
>
> **W3: Visualization on Mountain Car dataset**
>
> As the state space in mountain car is only two dimensional and the entire dataset is available it is easier to visualize the performance degradation for OOD states. The state space of sales promotion is 40000 (4 features for 10,000 users). Hence visualization of SP task is difficult. We show the performance improvement for Sales Promotion over baseline in Fig 3c Pg 7.
>
> **Definition 4.1**
>
> We define $\mathcal{B}_r$ as with respect to full dataset and not just states space because $\mathcal{B}_r$ also contains actions, rewards and next states. A performance drop is a result of both the conditions in Def 4.1 please refer to our analysis in Appendix B Pg 16.
>
> **Definition of $\eta$**
>
> Due to page limitations we provide the definition $\eta$ in Appendix A Pg 13. For any deterministic policy $\pi$ the performance return is formulated as $\eta(\pi) = E_{\tau \sim \pi}[\sum_{t=0}^{\infty} \gamma^t r(s_t, a_t)]$ following [3]. Where $\tau$ is trajectory, $\gamma$ is the discount factor and $r$ is the reward function. We will move the definition to the main text in revised manuscript.
>
> [3] Sham Kakade and John Langford.  Approximately optimal approximate reinforcement learning.  In 347 Proceedings of the Nineteenth International Conference on Machine Learning, pages 267–274, 2002
>
> **Definition of domain knowledge**
>
> Domain knowledge $\mathcal{D}$ is formally defined as hierarchical decision nodes capturing $S \to A$ where $A$ is non optimal as represented by Eq. 2 Pg 3. This can be generalized to any source of knowledge capturing $S \to A$ mapping for example dataset without reward or next state labels. General guidelines are typically available in practical domains like business, healthcare and autonomous driving which can be distilled in teacher network. _ExID will work with any type of domain knowledge that can be represented as S  →  A mapping._
>
> **Difference between the proposal and the traditional model distillation**
>
> Please refer to our response for W1.

---

> > ### Author Response · Authors · 2024-08-11
> > **Thank you for your feedback**
> >
> > Dear reviewer NyjZ,
> >
> > Thank you for your insightful review comments and for asking important questions that helped us improve the work. As the discussion period approaches towards end, kindly let us know if we have addressed your concerns and if there are feedbacks for discussion. Otherwise we will be grateful if you could reconsider your score.
> >
> > Thank you again for your efforts and we value your feedback deeply.
> >
> > Regards,
> > Authors

---

> > > ### Comment · Reviewer_NyjZ · 2024-08-13
> > >
> > > Thanks for your response which addressed my concerns. I will raise my score to 6.

---

> > > > ### Author Response · Authors · 2024-08-13
> > > > **Thank you for acknowledging the rebuttal**
> > > >
> > > > Dear reviewer NyjZ,
> > > >
> > > > Thanks for your kind support and for helping us improve the paper! We will incorporate the additional discussions in the revised version of the manuscript.
> > > >
> > > > Thank you again for your valuable comments and guidance.
> > > >
> > > > Best,
> > > >
> > > > Authors

---

> ### Comment · Area_Chair_woVH · 2024-08-12
> **Please confirm you've read the author response**
>
> Dear reviewer,
> Can you please confirm that you've read the author's responses?
> Given that the other review is now both vote for acceptance, it's important for you to voice any remaining concerns if you still believe the paper should not be accepted.
>
> Thank you!

---

### Official Review · Reviewer_Cy8u · 2024-07-13

**Soundness:** 4
**Presentation:** 4
**Contribution:** 4
**Rating:** 7
**Confidence:** 4

**Summary:**

The paper introduces a novel technique ExID, a domain knowledge-based regularization method, that adaptively refines initial domain knowledge to boost performance of offline reinforcement learning (RL) in limited-data scenarios. The key insight is leveraging a teacher policy, trained with domain knowledge, to guide the learning process of the offline-optimized RL agent (student policy). This mitigates the issue of erroneous actions in sparse samples and unobserved states by having the domain knowledge-induced teacher network to cover them. And the initial domain knowledge would be improved when the student policy reaches a better perform than the teacher policy.  Empirical evaluations on standard discrete environment datasets demonstrate a substantial average performance increase compared to traditional offline RL algorithms operating on limited data

**Strengths:**

1. Originality: The paper's originality lies in its integration of domain knowledge into offline RL through a teacher policy network. This approach addresses performance degradation in limited-data settings, which is a novel and underexplored area. The introduction of the domain knowledge-based regularization technique and adaptive refinement of initial domain knowledge are particularly innovative.

2. Quality: The quality of the work is evidenced by the solid theoretical analysis and the thorough empirical evaluations conducted on multiple standard datasets, including OpenAI Gym environments (Mountain Car, Cart-Pole, Lunar Lander) and MiniGrid environments, as well as a real-world sales promotion dataset. The results consistently show that ExID outperforms existing offline RL algorithms in these settings.

3. Clarity: The paper is well-structured, with clear explanations of the problem, methodology, and results. The use of diagrams and tables helps understand the motivation of the problem (figure 1), the proposed method (figure 2), illustrate the effectiveness of ExID (Table 1-2). Each section logically follows from the previous one, making the overall argument easy to follow.

4. Significance: By tackling the challenge of limited data in offline RL, the paper makes a significant contribution to the field. The proposed approach has practical implications for various real-world applications where data is scarce and expert knowledge is available, such as in business, healthcare, and robotics.

**Weaknesses:**

1. Generalization to Continuous Domains: The paper is limited to discrete action spaces, which restricts its applicability to a broader range of RL problems involving continuous action spaces. This limitation is acknowledged by the authors.


2. Scalability: The scalability of ExID to more complex environments that requires a complex representation (e.g., a significant large tree) of domain knowledge is not thoroughly explored.  It would be beneficial to understand how the method performs in such settings and what challenges might arise because the challenging of updating the domain knowledge represented in a complex representation could hinder the learning process of the student policy in the proposed method ExID.

**Questions:**

1. Comparative Analysis: Can the authors provide a more detailed comparison with other domain knowledge-based methods? Specifically, how does ExID differ in its approach to leveraging domain knowledge compared to methods like DKQ and CQL SE that
 are mentioned in Related Work?

2. Examples of Domain Knowledge Improvement: having some examples of the improved domain knowledge could make the paper stronger and more convincing.

**Limitations:**

The authors acknowledge several limitations of their work, including the reliance on the quality of domain knowledge and the focus on discrete action spaces. While these limitations are well-addressed in the paper, it may be worth to consider a broad evaluation:
   * Conducting experiments on a wider variety of environments that have larger state and action spaces, would provide a more comprehensive evaluation of the method's applicability.

---

> ### Author Rebuttal · Authors · 2024-08-06
>
> Thank you for your appreciative and constructive feedback. Please find our response below:
>
> **W1 Generalization to Continuous Domain**
>
> - The proposed methodology can be extended to any continuous domain problem by using the regularization in Eq 4 : $\mathcal{L}(\theta) = \mathcal{L}cql(\theta)  + \lambda
> E_{s \sim \mathcal{B}r \land s \models \mathcal{D}} [Q_s^\theta(s, a_s)-Q_s^\theta(s,a_t)]^2$ during critic $(Q_s^\theta)$ training for continuous domain and using actions from actor network $(\pi_s)$ for cross entropy loss in Eq 7 : $\mathcal{L}(\omega) = -\sum_{s \models D} (\pi_t^\omega(s)log(\pi_s(s)))$.
> - We empirically demonstrate this by conducting additional experiment by training policy on continuous version of sales promotion task (action range coupon (0 - 5), discount (0.6 - 1) ) and  continuous Type-1 Diabetes basal bolus control task (action range basal and bolus (-1 to 1) for SimGlucose dataset. Please refer to the results in table below:
>
> | Environment |CQL+$\mathcal{D}$ |SCQ |EXID|Performance gain|
> |--|--|--|--|--|
> | SP|679.25$\pm$ 35.02 | 708.44 $\pm$ 52.19 |827.76 $\pm$ 43.79  | 14.38% |
>
>
> | Environment |$\mathcal{D}$| CQL+$\mathcal{D}$ |EXID|Performance gain|
> |--|--|--|--|--|
> | SimGlucose |17.53 $\pm$ 3.02 | 21.79 $\pm$ 3.60 |30.82 $\pm$ 6.95  | 41.44% |
>
> **W2 Scalability**
>
> EXID can be scaled for complex domain knowledge trees if the teacher neural network is able to capture the $S \to A$ mapping of the tree. However the effect of complexity on the teacher network is currently beyond the scope of this paper.
>
> **Q1 Comparative Analysis**
>
> - DKQ uses Q operator guided by domain knowledge using Eq: $\mathcal{T}{\mathcal{F}} Q(s, a) := r(s, a) + \gamma E{s' \sim P(s'|s, a)} \left[ \sum_{i=1}^{K} \alpha_i \max_{a' \in \text{supp}(f_i)} Q(s', a') \right]$
> which requires the importance of all actions $a' \in \text{supp}(f_i)$ to be labelled from domain knowledge. The domain knowledge is not updated during the training process. This operator also only works on the states observed in the dataset. Contrary to this our method does not require action support labels for each action and incorporates the domain knowledge for unseen states through knowledge distillation and teacher update.
> - CQL SE uses an uncertainty-weighted regularization of OOD actions using safety experts represented by $Q(s, a) = r + \gamma \ast \max_{a'} Q(s', a') - \underbrace{(1 - conf(s)) \ast (a - \pi_T(s))^2}_{\text{uncertainty weighted learning from the safety expert}}$. The safety expert is considered to be optimal and not updated during training. conf(s) is calculated based on states observed in the dataset and this method does not account for unseen states.
>
> **Q2 Examples of Domain Knowledge Improvement**
>
> Since the domain knowledge is partial, direct improvement of $\pi_t^\omega$ is not always observed. We show domain knowledge improvement using ablation study in Fig 5c which depicts no teacher update leads to a suboptimal policy. This is also depicted using baseline CQL+D where simply using domain knowledge without improving it does not lead to an optimal performance.

---

> > ### Comment · Reviewer_Cy8u · 2024-08-11
> >
> > I thank the authors for the detailed response. My concerns have been addressed. And I would like to remain my score.

---

> > > ### Author Response · Authors · 2024-08-11
> > > **Thank you for acknowledging the rebuttal**
> > >
> > > Dear Reviewer Cy8u,
> > >
> > > Thanks for your kind support and for helping us improve the paper! We will incorporate the additional discussions in the revised version of the manuscript.
> > >
> > > Thank you again for your valuable comments and guidance.
> > >
> > > Best,
> > >
> > > Authors

---

### Author Rebuttal · Authors · 2024-08-06

We thank the reviewers for their time and constructive feedback and for highlighting the following strengths.

**Strengths** – Novel and unexplored (Reviewer Cy8u), solid theoretical analysis (Reviewer Cy8u),  simple and technically reasonable (Reviewer Cy8u, Nyzj), Practically applicable (Reviewer Cy8u, NyjZ), Domain Knowledge Utilization (Reviewer UWfE).

We acknowledge the areas of improvement the reviews have suggested and have made concerted efforts to address them. We would like to highlight our response to the major concerns and then respond to the individual reviews.

**Discrete Action Space Limitation and more real world experiments**

- We conducted experiments on discrete action space domains as many important real world problems use discrete action policies. For example navigation with actions forward or turning, finance trading with actions buy, hold or sell etc. We have also mentioned the use of discrete policies in our limitations.
- However, we will like to highlight the proposed methodology can be extended to any continuous domain problem by using the regularization in Eq 4 :  $\mathcal{L}(\theta) = \mathcal{L}cql(\theta)  + \lambda
E_{s \sim \mathcal{B}r \land s \models \mathcal{D}} [Q_s^\theta(s, a_s)-Q_s^\theta(s,a_t)]^2$ during critic $(Q_s^\theta)$ training for continuous domain and using actions from actor network $(\pi_s)$ for cross entropy loss in Eq 7 : $\mathcal{L}(\omega) = -\sum_{s \models D} (\pi_t^\omega(s)log(\pi_s(s)))$.

- We empirically demonstrate this by conducting additional experiment by training policy on continuous version of sales promotion task (action range coupon (0 - 5), discount (0.6 - 1) ) and  continuous Type-1 Diabetes basal bolus control task (action range basal and bolus (-1 to 1) for SimGlucose [2] dataset. Please refer to the results in table below:

| Environment |CQL+$\mathcal{D}$ |SCQ [1] |EXID|Performance gain|
|--|--|--|--|--|
| SP|679.25$\pm$ 35.02 | 708.44 $\pm$ 52.19 |827.76 $\pm$ 43.79  | 14.38% |

All the experiment plots conducted during rebuttal are provided in attached pdf.

| Environment |$\mathcal{D}$| CQL+$\mathcal{D}$ |EXID|Performance gain|
|--|--|--|--|--|
| SimGlucose |17.53 $\pm$ 3.02 | 21.79 $\pm$ 3.60 |30.82 $\pm$ 6.95  | 41.44% |

- In the manuscript we have shown empirical results on 6 benchmark opensource datasets and theoretically establish in proposition 4.2 the use of imperfect domain knowledge can lead to performance improvement in offline RL policies.

For the rebuttal phase we have done the following additional experiments as suggested by reviewers

1. Empirical study on continuous sales promotion and Type 1 diabetes task to address discrete action space limitation raised by reviewer UWfE and reviewer Cy8u.
2. Empirical comparison with baseline SCQ [1] on Sales Promotion task for strong experiment comparison as suggested by reviewer UWfE.
3. Experiment on Type-1 Diabetes basal bolus control dataset for other real world experiments as suggested by reviewers UWfE and NyjZ


[1] Shimizu, Yutaka, et al. "Strategically Conservative Q-Learning." arXiv preprint arXiv:2406.04534 (2024).

[2]. Jinyu Xie. Simglucose v0.2.1 (2018) [Online].

---

### Decision · Program_Chairs · 2024-09-25

**Decision:**

Reject

**Comment:**

Reviewers agreed that this paper should be accepted, and praise the approach as simple, reasonable, and innovative.  However, I noticed that the paper does not mention model-based offline RL or include any model-based offline RL approaches as baselines.  I consider this a critically important ommission, since:
* The current related work mischaracterizes the field of offline RL, stating "Two techniques have been studied in the literature to prevent such errors. 1) Constraining the policy to be close to the behavior policy 2) Penalizing overly optimistic Q values [24]."  However, model-based approaches are distinct from these, as noted in [24]: "Conceptually, these methods resemble the conservative estimation approach described in Section 4.5, but instead of regularizing a value function, they modify the MDP model learned from data to induce conservative behavior."
* Model-based offline RL methods such as MOPO and MoReL were found to outperform approaches (1) and (2) over 4 years ago.  The papers introducing those methods were published at NeurIPS 2020 and are more highly cited than most of the works the authors reference in their related work section.
* Concurrently with MOPO and MoReL, in "The Importance of Pessimism in Fixed-Dataset Policy Optimization", Buckman et al. noted theoretical reasons why existing approaches (with the exception of the model-based approaches) were fundamentally limited.

This makes me seriously question the strength of the baselines that are used in this work and the authors situation of their work within the literature, and reviewers were unable to reassure me on these points.  Thus I am unable to recommend acceptance at this point.